# The Impact of Tumor Edema on T2-Weighted 3T-MRI Invasive Breast Cancer Histological Characterization: A Pilot Radiomics Study

**DOI:** 10.3390/cancers13184635

**Published:** 2021-09-15

**Authors:** Domiziana Santucci, Eliodoro Faiella, Ermanno Cordelli, Alessandro Calabrese, Roberta Landi, Carlo de Felice, Bruno Beomonte Zobel, Rosario Francesco Grasso, Giulio Iannello, Paolo Soda

**Affiliations:** 1Department of Radiology, University of Rome “Campus Bio-Medico”, Via Alvaro del Portillo, 21-00128 Roma, Italy; e.faiella@unicampus.it (E.F.); roby_landi@libero.it (R.L.); b.zobel@unicampus.it (B.B.Z.); r.grasso@unicampus.it (R.F.G.); 2Unit of Computer Systems and Bioinformatics, Department of Engineering, University of Rome “Campus Bio-Medico”, Via Alvaro del Portillo, 21-00128 Roma, Italy; e.cordelli@unicampus.it (E.C.); g.iannello@unicampus.it (G.I.); p.soda@unicampus.it (P.S.); 3Department of Radiology, University of Rome “Sapienza”, Viale del Policlinico, 155-00161 Roma, Italy; alessandro.calabrese.92@gmail.com (A.C.); c.df@uniroma1.it (C.d.F.)

**Keywords:** breast cancer, 3T-MRI, lymph node status, machine learning, radiomics, signature

## Abstract

**Simple Summary:**

Breast cancer is the most common cancer in women worldwide. Currently the use of MR is mandatory in staging phase. The standard protocol includes T2-weighted sequences for morphology and signal analysis, T1-weighted images for adding information (i.e., ematic or adipous components), diffusion-weighted sequences which provide information on tissue cellularity, and dynamic post-contrast sequences useful for detecting and locating lesions. Although not considered among the main prognostic factors in current guidelines, tumor-associated edema provides useful information on tumor aggressiveness, and has been shown to be associated with the main histological tumor characteristics. With this work, entitled “The Impact of Tumor Edema on T2-weighted 3T-MRI Invasive Breast Cancer Histological Characterization: a Pilot Radiomics Study”, we want to demonstrate that radiomics edema, based on algorithms that allow the extraction of imaging features not visible to the human eye, can further increase the accuracy in the prediction of histological factors compared to the use of traditional information only.

**Abstract:**

Background: to evaluate the contribution of edema associated with histological features to the prediction of breast cancer (BC) prognosis using T2-weighted MRI radiomics. Methods: 160 patients who underwent staging 3T-MRI from January 2015 to January 2019, with 164 histologically proven invasive BC lesions, were retrospectively reviewed. Patient data (age, menopausal status, family history, hormone therapy), tumor MRI-features (location, margins, enhancement) and histological features (histological type, grading, ER, PgR, HER2, Ki-67 index) were collected. Of the 160 MRI exams, 120 were considered eligible, corresponding to 127 lesions. T2-MRI were used to identify edema, which was classified in four groups: peritumoral, pre-pectoral, subcutaneous, or diffuse. A semi-automatic segmentation of the edema was performed for each lesion, using 3D Slicer open-source software. Main radiomics features were extracted and selected using a wrapper selection method. A Random Forest type classifier was trained to measure the performance of predicting histological factors using semantic features (patient data and MRI features) alone and semantic features associated with edema radiomics features. Results: edema was absent in 37 lesions and present in 127 (62 peritumoral, 26 pre-pectoral, 16 subcutaneous, 23 diffuse). The AUC-classifier obtained by associating edema radiomics with semantic features was always higher compared to the AUC-classifier obtained from semantic features alone, for all five histological classes prediction (0.645 vs. 0.520 for histological type, 0.789 vs. 0.590 for grading, 0.487 vs. 0.466 for ER, 0.659 vs. 0.546 for PgR, and 0.62 vs. 0.573 for Ki67). Conclusions: radiomic features extracted from tumor edema contribute significantly to predicting tumor histology, increasing the accuracy obtained from the combination of patient clinical characteristics and breast imaging data.

## 1. Introduction

In 2020, breast cancer (BC) was the most commonly diagnosed cancer with 2.3 million new cases worldwide (11.7%) [1].

BC is considered a heterogeneous disease with different phenotypes, histological types, and molecular subtypes [2,3]. It is important to correctly define BC characteristics, as all these distinct factors may lead to different outcomes for each patient.

Histological classification and imaging stadiation are imperative for guiding breast cancer patient management. Histologic prognostic factors, which predict the risk of death and recurrence from BC, can be morphological, including tumor histology, tumor nuclear grade, lymphatic and vascular invasion, and molecular, such as estrogen-receptor (ER) and progesterone-receptor (PgR) status, epithelial growth factor receptor HER2 status, and expression of proliferation-related genes, such as Ki-67 [4,5,6,7].

Breast edema is not considered a major prognostic factor in current national and international guidelines (ACR and AIOM) [8,9]. However, several studies have shown a correlation between breast edema and tumor aggressiveness [10,11,12,13,14]. Focal edema is mostly correlated with malignancy, representing an indirect sign of peritumoral vascular invasion and inflammation.

As a highly sensitive and non-invasive method, magnetic resonance imaging (MRI) has a prominent role in BC diagnosis, pre-operative tumor staging and patient management, providing both morphologic and functional information [15].

However, MRI tumor information is qualitative, and results depend on the radiologist. Among the different MRI-sequences, the post-contrast sequences are used for the characterization of lesion behavior, while pre-contrast images give mainly morphologic information [15,16]. Focal edema appears as high signal intensity on T2-weighted (T2-WI) MRI sequences and, based on its location, can be classified in three different types: peritumoral, prepectoral, and subcutaneous edema [10].

Artificial Intelligence (AI) is increasingly being used in oncology for improved decision support. Radiomics is a branch of AI that involves converting information contained in routinely collected medical images into quantifiable and measurable data, also referred to as “features” [17,18]. In BC, radiomics is applied as a decision supporting tool, potentially improving lesion characterization and patient prognosis, mostly using MRI [19]. In recent literature several studies have evaluated the correlation between tumor features, extracted from MRI, and breast cancer molecular subtypes, with promising results [20,21,22,23,24,25,26,27,28,29]. However, to our knowledge, there are no articles associating tumor edema radiomics with primary tumor imaging information in order to predict breast cancer biological aggressiveness.

Our aim is, therefore, to evaluate the contribution of edema associated with clinical and MRI features for breast cancer histological prognostic factors prediction, using T2-weighted MRI radiomics.

## 2. Materials and Methods

### 2.1. Study Population

In this study all breast cancer MRI examinations performed at the Department of Radiological Sciences, Sapienza University in Rome, for local staging, from January 2015 to January 2019, were retrospectively reviewed. A total of 160 patients, with 164 histologically proven invasive BC lesions, were enrolled. 

The following were considered as inclusion criteria: staging 3T MRI examination; presence of Dynamic contrast-enhanced-MRI (DCE) T2-WI and DWI sequences; presence of focal or diffuse edema in the T2-WI sequences, histopathologic diagnosis of invasive BC; complete histologic analysis, including molecular receptor evaluation (estrogen receptor ER, progesterone receptor PgR; epidermal growth factor receptor HER2) and Ki-67 index calculation.

Exclusion criteria were: presence of breast implants, post-chemotherapy follow-up patients, neo-adjuvant chemotherapy, and images that were not of good diagnostic quality. 

Patients clinical data (age, menopausal state, family history, hormone therapy), tumor MRI-features (location, margins, dimensions, morphology, kinetic curves, edema type), and histological features (histological type, grading, ER, PR, HER2, and Ki-67 index) were collected.

Lymph node tumoral involvement for each patient was also recorded, using definitive surgical characterization, as a dichotomous result: positive, if there was at least sentinel LN involved by BC metastasis, or negative, if there was no lymph node metastasis.

Patients’ clinical data and tumor MRI-features were used as “semantic features” while the histological factors were used as labels to be predicted for the algorithm.

### 2.2. MRI Examination

All MRI exams were performed on a 3T magnet (Discovery 750; GE Healthcare, Milwaukee, WI, USA). Patients were positioned prone and a dedicated eight-channel breast coil (8US TORSOPA) was used. Three orthogonal localizer sequences were employed, then images were acquired following this protocol:T2-weighted axial single-shot fast spin echo sequence with a modified Dixon technique (IDEAL) for intravoxel fat-water separation (TR/TE 3500–5200/120–135 ms, matrix 352 × 224, FoV 370 × 370, NEX 1, slice thickness 3.5 mm). Diffusion weighted axial single-shot echo-planar with fat suppression sequence (TR/TE 2700/58 ms, matrix 100 × 120, FOV 360 × 360, NEX 6, slice thickness 5 mm) with diffusion-sensitizing gradient applied along the three orthogonal axes and with a b-value of 0, 500, and 1000 s/mm^2^.T1-weighted axial 3D dynamic gradient echo sequence with fat suppression (VIBRANT) (TR/TE 6.6/4.3 ms, flip angle 10°, matrix 512 × 256, NEX 1, slice thickness 2.4 mm), before and five times after intravenous contrast medium injection.Current guidelines suggest at least three time points to measure during the post-contrast-phase: one before the administration of contrast medium, one approximately 2 min later to capture the peak, one in the late phase. This allows us to evaluate whether a lesion continues to enhance or is characterized by contrast agent wash-out. At least two measurements after contrast medium administration are recommended, even if the optimal number of repetitions is unknown. In our center, we usually perform five acquisitions after contrast medium administration ensuring to obtain a specific signal intensity curve time without penalizing the duration of the examination.

Gadobenate-dimeglumine (Multihance^®^; Bracco Imaging, Milan, Italy) was administered at a concentration of 0.2 mmol/kg and injected intravenously (20 G cannula) at a rate of 2 mL/s via an automatic injector; this was followed by infusion of 15 mL of saline at the same rate. In post-processing, subtracted images were automatically produced from the images after contrast medium administration for a more accurate tumor analysis.

For each lesion the following MRI characteristics were collected using DCE-sequences:-Location on the breast quadrant; -Margins: regular, irregular, lobulated, spiculated or non-mass; -Dimensions (mm); -Morphology: round, oval, or irregular;-Contrast enhancement, quantified using the signal intensity/time curve: type I, characterized by a slow wash-in and without wash-out, type II, defined by a plateau curve after a rapid/slow wash-in, and type III, with rapid wash-in and rapid wash-out;-Associated-tumor edema type.

### 2.3. Edema Evaluation

The 164 lesions were divided according to the presence or absence of edema (0). When present, edema was classified as peritumoral (1), pre-pectoral (2), subcutaneous (3), or diffuse (4). Two example of edema types are reported in Figure 1. 

Only lesions associated with edema were eligible for this study. Edema was identified as high signal intensity on T2-weighted sequences. However, edema may have relatively similar signal to tumor lesions and breast tissue in the conventional T1-weighted images and the T2-fat-suppression sequences. Therefore, the DCE and DWI sequences were also used, allowing the best definition between tumor and edema. In addition, a comparison between the two breasts was exploited in order to distinguish breast edema, in the breast side with tumor, from breast glandular tissue, in the breast side without tumor.

### 2.4. Histological Characteristics

Breast tissue specimens obtained after biopsy or surgery were analyzed by a pathologist with more than 20 years of experience. Histological diagnosis was performed following the WHO classification: the histopathological grade was evaluated according to NGS (Nottingham Grading Score), through a scoring system which evaluate tubule formation, pleomorphism, and mitotic count. The total numerical score ranges from 3 to 9. A score of 3–5 corresponds to grade 1 (G1), a score of 6 or 7 was interpreted as grade 2 (G2), and a total score of 8 or 9 leads to diagnosis of grade 3 (G3).

Immunohistochemical (IHC) analysis was performed to evaluate molecular receptors status (ER, PgR, and HER2) and to calculate Ki-67 index. Only nuclear reactivity was considered for ER. The monoclonal antibody Mib-1 (1:200 dilution; Dako, Glostrup, Denmark) was used to assess the Ki-67 index, which was reported as the percentage of immune-reactive cells out of 2000 tumor cells in randomly selected high-power fields surrounding the tumor core. HER2 status was re-evaluated using the Hercep test (Dako, Glostrup, Denmark). Samples that gave an equivocal IHC result were subjected to fluorescence in situ hybridization (FISH) analysis. A ratio of HER2 gene signals to chromosome 17 signals greater than 2.2 was used as a cut-off value to define HER2 gene amplification. ER and PgR status was considered positive if expression was ≥1% and negative if expression was <1%. HER2 expression was graded as follows: 0, 1+, 2+, or 3+; tumors with a score greater or equal than 2+ were considered HER2 positive, whereas scores lesser than 2+ were considered HER2 negative. Ki-67 expression ≥14% was considered positive and <14% was considered negative.

### 2.5. Statistical Analysis

Descriptive statistics were accomplished using the statistical software program SPSS© version 25.0. Statistical significance was set at *p* < 0.05.

Spearman’s Rank-Order Correlation was evaluated to assess whether there was a correlation between the edema types and categorical variables, both clinical (menopausal status, hormone therapy, family history), MRI (location, margins, dimensions, morphology, kinetic curves, edema type), histological characteristics (histologic type, grading, expression of ER, PgR, and HER2b and Ki-67 index) and lymph node status (positive or negative). We also performed Spearman’s Rank-Order Correlation to assess a possible correlation between edema types and positive lymph node status, excluding lesions with peritumoral edema.

The Kolmogorov-Smirnov test was performed to determine whether age and tumor size followed a normal distribution. 

Statistical comparison of edema type and age and tumor size was performed using the Kruskal-Wallis H test. 

### 2.6. Radiomics

#### 2.6.1. Segmentation

Each case has been anonymized and identified with a progressive identification number (ID). For the analysis of bilateral tumors, lesions were considered one at a time with different IDs. For each breast sample the T2-weighted sequence was selected and loaded into a workstation. To get the region of interests (ROIs), a semi-automatic segmentation of the edema was performed by an expert radiologist, and then proofread by another one, with 3D Slicer (http://www.slicer.org, accessed date: 1 January 2015). All the following analysis were run using home-made algorithms on MATLAB^1^. Segmentation was performed at first in the axial projections, avoiding tumor lesions and necrosis when present. Each segmentation was improved in the sagittal and coronal projections and finally optimized using three-dimensional version of the well-known convex-hull algorithm. 

Indeed, although all tumor edemas usually present a spherical and irregular geometry, when considering an MR acquisition it is easy to incur in some tissues that are partially difficult to segment, either because of tissue stretching due to the imaging phase, or because of hard-to-detect shapes within the images when edemas are close to other tissues with similar density, such as cysts or pure dense glandular tissue, thus often resulting in jagged edges and sometimes concavities in the resulting ROI. On this ground, the calculation of the three-dimensional convex hull serves a twofold purpose: firstly, it removes all possible segmentation deformities otherwise hindering further calculation of textural features and secondly, it can slightly expand the ROIs thus also containing a perimeter portion of healthy tissue at the edge of the segmentation that might actually carry information about the tumor status.

#### 2.6.2. Feature Extraction and Selection

A total of 253 features were extracted and filtered, selecting the most informative through a Best Firsts wrapper algorithm, and using a Random Forest and the labels we would like to predict. We performed five experiments, by using the different histological labels set with dichotomic values (0–1): histology, grading, ER, PgR, and Ki-67. All the experiments will be detailed in the next section.

Furthermore, it is worth noting that the wrapper-based feature selection was applied avoiding any bias between the training and test set and we used all default Matlab’s parameters for the classification.

The classification stage straightforwardly employed the same learning paradigm used in the feature selection phase; note also that the use of a Random Forest alleviates the curse of dimensionality and permits us to benefit by the aggregation of several decision trees, mimicking the consensus mechanism among different experts. This configuration was applied using only semantic features (i.e., patient clinical data and features derived by visual inspection of the MR images), and semantic features associated to edema radiomics descriptors.

We preferred to manually implement the code in order to better control specific computational aspects, usually bulked into external softwares, considered when generalizing features generally extracted on planar images to a three-dimensional volume. All the steps to build our code were tested against well-established packages.

A total of 253 features were extracted, of which 11 were semantic (age, menopausal status, family history, hormone therapy, MRI location, stadiation, margins, dimensions, morphology, kinetic curves, edema type) and 241 were radiomics. In order to fully take advantage of gathered acquisitions, no data nor gray level reduction was performed to the images and all analysis were intrinsically computed in the three-dimensional voxel space. It worth noting that although the relatively large slice thickness of the MRs could alter the isotropy of the acquired stacks, all the features we extracted which are sensitive to the voxels texture orientation were computed for all the existing three-dimensional orientations and then filtered out, in the further feature selection step, all the less informative ones. In detail, there were 11 first-order features (mean, standard deviation, skewness, kurtosis, energy, entropy, maximum absolute value, position of the maximum absolute value, energy around the maximum absolute value, range, number of maximum relative values, energy around the maximum relative values), 48 s-order features extracted from the histograms of four variations of the three Orthogonal Planes-Local Binary Patterns (TOP-LBP) (the same feature extracted from the first-order group, per each variation), and 182 features extracted from the 3D version of the Gray Level Co-occurrence Matrix (GLCM) (autocorrelation, covariance, inertia, absolute of the inertia, inverse of the inertia, energy and entropy, per each of the 26 possible three-dimensional directions). A brief summary of such features is reported in Table 1.

#### 2.6.3. Classification

As mentioned before, the labels have always been assumed dichotomic values of 0 and 1: for histology (ductal or lobular invasive carcinoma), for ER and PgR status (≥ or <10%), for Ki67 proliferation index (≥ or <1%), and also for grading (G1 + G2 vs. G3), grouping the class with the lowest number of elements (G1 in the specific case). The dichotomic labeling approach was chose in order to avoid a class unbalance and to ease and standardize the classifier approach.

Each of the five experiments was performed in ten-fold cross-validation, one at histological label to predict. The process details are showed in the Figure 2.

Finally, to quantify the results and evaluate the overall classification performance, the prediction Confusion Matrices were obtained, and both ROC area (AUC) and accuracy (ACC) were considered as evaluation metrics for each label.

## 3. Results

In this study, 164 breast cancer lesions with 127 presenting edema tumor were enrolled. Four patients had bilateral breast cancer, but only one patient with bilateral BC presented edema. Edema types were represented as follows: 62 (49%) peritumoral, 26 (20%) pre-pectoral, 16 (13%) subcutaneous, 23 (18%) diffuse. The history-clinical data of the study population were collected. The mean age of the patients presenting edema was 54.86 years (range 30–84). Seventy-two (*n* = 72, 45.5%) patients were pre-menopausal and 88 (55.5%) were postmenopausal; 120 (73%) patients did not have any family member affected by breast cancer, 32 (20%) patients had one family member, and 8 (5%) patients had at least 2 female or male family members affected by breast cancer at any age.; 9 (6%) patients assumed hormone therapy for at least 3 continuous months, whether for contraceptive, substitution, or medical therapy reasons, and 151 (94%) patients did not assume any hormone therapy.

The median diameter of the lesions measured was 19 mm (range 9–60 mm).

Analyzing tumor location, peritumoral edema was more present when the tumor was located at the upper-outer quadrant (30.6%), pre-pectoral was more frequent when the tumor was in the internal quadrant (upper 12% and lower 46%), subcutaneous edema was predominant for lesions located at the lower-inner quadrant (21%), while the diffuse class was more frequent in case of central tumors (25%). 

The lesions were more frequently round or oval in peritumoral edema (56%), irregular in case of subcutaneous and pre-pectoral edema (36.4% and 41.8% respectively) and non-mass like in case of diffuse edema (30.6%).

The other main clinical, imaging, and histological characteristics of the 126 patients presenting edema, divided basing on edema type, are shown in Table 2.

Using Spearman’s Rank test no correlation was found between edema type and menopausal status, hormone therapy, lesion margins, kinetic curve, histological type, and positive lymph node status (*p* value > 0.5). A significant correlation was found between tumor histologic class, grading, ER, PgR and HERb2 expression, and Ki-67 index. A significant correlation was found between edema type and positive lymph node status, when peritumoral edema was excluded (*p* = 0.022). The Kruskal-Wallis H test demonstrated a significant correlation between edema class and lesion size (*p* value < 0.001). No correlation was found between patient age and edema type.

### Radiomics Approach

After the feature extraction, a signature, composed by the most significative features that lead to the final results, was individuated for each histological label (histological class, ER and PgR status, Ki-67 index and grading). The selected features and their predictive impact for each histological label were reported in the following charts (Figure 3).

For the sake of completeness, we would like to stress the fact that in order not to incur in the issue known as Curse of Dimensionality, i.e., when during a classification task the number of features is empirically higher than 10% of the samples present in the dataset, thus biasing the predictions, after the feature selection phase we should have concluded with at least 10 edemas in the minority class per feature, and instead the number of selected features exceeds the number of analyzed samples in all the classes considered. In fact, after the feature selection we accepted a certain abundance of extra features without proceeding with a deeper filtering, due to the fact that the algorithm considered in the next step, the Random Forest, is quite capable of dealing with a larger number of attributes than the relevant ones, performing an additional intrinsic selection of only the most significant features and avoiding bias.

The difference between the AUC and the accuracy obtained to predict each of the five histological prognostic labels, with and without edema radiomics features contribution, is reported in Table 3.

## 4. Discussion

Among the different imaging methods, MRI has demonstrated an increasingly important role as a breast cancer management guide, from BC diagnosis and loco-regional stadiation, to aiding in therapy decision-making [8,30]. High-field MRI contrast resolution allows a very accurate definition of the main morphological and functional tumor characteristics.

In recent years, in the literature there have been preliminary studies that have attempted to seek a correlation between MRI tumor characteristics with BC molecular subtypes and prognostic factors using artificial intelligence (AI). Radiomics, which represents the main AI medical application, consists of the analysis of medical images and aims to convert the relative information and characteristics, called features, that the observer is not able to see by themselves, into quantitative and measurable high-data functions.

MRI post-contrast sequences have proved to be the main sequences for the phenotypic description of the lesion, characterization of the true tumor extension, margins, morphology and, almost importantly, definition of tumor neoangiogenesis [31]. On the other hand, the pre-contrast sequences, T2-weighted and DWI, are the source of morphological information and are used to improve tumor aggression characterization, contributing with data about the degree of peri-tumor inflammation.

Edema, identified as a high-intensity signal in T2-weighted sequences, can therefore be useful, although non-specific, data, correlating with peritumoral malignant lesion spread and lymphangioinvasiveness [10,11,12,32,33]. Edema can be classified into peritumoral, pre-pectoral, subcutaneous, and diffuse, presenting a different degree of correlation with tumor aggressiveness. Both malignant and benign lesions, such as mastitis, previous radiotherapy, post-surgical inflammation, nephrotic syndrome, lymphoma, venous congestion, and chronic heart disease can be a cause of diffuse breast edema [34]. Focal edema, studied in T2WI sequences, however, is associated with malignant lesions in most cases [10,13,35,36,37,38].

Peritumoral edema is caused by the tumoral process of neoangiogenesis, resulting in increased vascular permeability in the newly formed vessels and the release of peritumoral cytokines. [14]. This is specific of invasive breast cancer, even if it is less frequent in lobular cancer than in ductal cancer [39]. This could be due to the low density of the lesion and the growth pattern of lobular carcinoma. Peritumoral edema is also significantly related to rim enhancement [40].

Breast tissue is drained primarily by lymphatic vessels leading to the axillary lymph nodes [41,42]. Invasive BC causes lymphatic vessels to increase in number and size, consequently increasing the risk of lymph node metastasis [43]. When the main axillary lymphatic drainage is blocked due to carcinoma, collateral intramammary and pre-pectoral lymphatic drains take over [10,32,37,41,42,44]. Pre-pectoral edema is related to tumor cells in the retro-mammary areas [30,37,45]. When lymphatic drainage in the subcutis is blocked due to tumor emboli, subcutaneous edema will occur [37,44]. In theory, pre-pectoral edema usually precedes subcutaneous edema as it is considered by Uematsu et al. as “the final stage of breast edema associated with malignancy”. During imaging studying, it is therefore useful to clarify the type and the quantity of edema observed, as this finding can help in predicting the malignancy of the lesion.

On this basis, we wanted to explore edema over-reader information, with the aim to analyze features which can be added as imaging breast cancer descriptors to the traditional radiologist evaluation, by combining information from radiomics, based on T2-weighted 3T MRI sequences, with the patient clinical and traditional imaging data of the lesion to predict the histological characteristics of the tumor.

To our knowledge there are no articles that associate tumor edema radiomics to tumor imaging in order to improve breast cancer biological aggressiveness.

In our study, the class of edema correlates significantly with tumor size, histological class, grading and expression of ER, PgR and HER2b, and Ki-67 index, reflecting how the presence of edema is associated with particularly aggressive tumors. As already hypothesized by Baltzer et al. invasive tumor growth and progression are associated with phenomena of tumor proteolysis and neoangiogenesis; these factors contribute to an increased vascular permeability, because of basal membranes that perform less than in the physiological vasculature [35]. Tumor growth and progression are nothing but the expression of tumor aggressiveness, which is in turn expressed at histological level by the nuclear grade, mitotic growth index, and receptor expression of the tumor lesion.

We did not find a significant correlation between the class of edema and the presence of lymphadenopathy, when considering all the edema types. When considering the lesions with pre-pectoral, subcutaneous, and diffuse edema, a significant correlation was found, confirming Uematsu’s studies. Uematsu describes the classes of pre-pectoral and subcutaneous edema as signs of lymphovascular invasion (LIV), which is responsible for disruption of lymphatic drainage in the dermal and subdermal area due to tumor emboli: LIV is significantly associated with pre-pectoral edema, and subcutaneous edema which is its final stage, and both are associated with inflammatory breast cancer and occult inflammatory breast cancer [10].

Regarding the radiomics impact, our results show consistently higher AUC using breast edema radiomics features. The best improvement in AUC was obtained regarding grading (0.789 vs. 0.590) while estrogen- and progesterone-receptors showed a lower increase of AUC (ER: 0.487 vs. 0.466; ER: 0.659 vs. 0.546). The improvements can be confirmed by looking at the values of sensitivity, specificity, PPV, and NPV of each experiment. In fact, the percentages almost always show higher values when considering predictions obtained using edemas when calculating features rather than not, only in a few comparisons we have that specificity shows a lower result as well as a single NPV, suggesting a more conservative trend for the classifier using edema features. These results can probably be explained as edema is an imaging manifestation of advanced or inflammatory BC, while hormone-receptors are histological features. The only case in this study where the accuracy decreases is in case of histological type. This can be explained by the imbalance of our sample (108 CDI vs. 17 CLI).

It is also interesting to note that only some semantic features survived to the features selection process, such as the DCE-Kinetic Curve, which represents the tumor intrinsic vascularization, closely connected with the lesion aggressiveness and edema, as strongly reported in literature [10,11,12,13,14,15,16,17,18,19,20,21,22,23,24,25,26,27,28,29]. However, for all the histological labels, the majority of selected features were radiomics, and in particular second-order ones. This result suggests that, for this experiment, most of the information resides within the tissue pattern and the micro-structure of the tumor contours, which are the properties analyzed by these feature families. The spread of tumor-associated edema follows specific patterns related to tumor aggressiveness, and this can explain this class features selection.

The use of a 3Tesla magnet allows the generation of high-quality images with high information content. Our study collected images with high data content to be exploited to obtain features that are not only numerous, but also highly detailed in order to achieve the maximum correlation with the histological microstructure and facilitate the radiomic path. This is possible based on the greater detail provided by high-resolution images, as demonstrated by various studies performed on high-field MRI [46,47].

High resolution image production guarantees an excellent definition of the details allowing a selection of most appropriate features, but also optimizes the work of the classifier, both from a computational point of view, streamlining the amount of information to be evaluated, and the result, ensuring greater appropriateness of the data.

It has emerged over the years that the application of Artificial Intelligence, even in other fields, requires very advanced computational sciences and statistics. Moreover, radiomics needs an incredibly collaborative inter-disciplinary, inter-institutional team [48].

The main limitations of this study include the small sample size and the single center nature of the study, the unbalanced number of the group component (smaller number of the minority class), the manual segmentation, leading to time-consumption and user error and variability, the large number of features utilized which may lead to overparameterization, the lack of an independent test set, and the lack of an assessment of inter-reader variation. A larger and preferably multi-center cohort is needed for a more rigorous analysis. Moreover, an automatic, standardized, and validated segmentation method would be ideal even if not yet available at present. It is certainly necessary to keep in mind the inherent limits of radiomics: this technique is currently still tied to the radiologist’s ability to select ROIs and the engineer’s ability to build the algorithm capable of feature selection and their experience.

However, the results obtained in this preliminary study, as well as those published so far in the literature, are highly promising. Considering the significant clinical impact that the appropriate use of these tools could offer the physician, the growing research in this area is justified. There is the need to produce common datasets in the public domain, in order to make the collected data usable and to standardize the protocols.

## 5. Conclusions

This study demonstrates that the BC edema radiomic features of preoperative 3 Tesla MRI extracted from the T2 sequences significantly correlate with the main histological prognostic factors. Despite its currently marginal role in guidelines as a predictor, the breast associated edema may be useful in the preliminary phase to provide additional information on tumor bio-histological aggressiveness. This concept is more evident by applying Artificial Intelligence algorithms to images that can provide objective quantitative data to the limited vision of the radiologist. Radiomics is a complex tool in its implementation phase but simple in its use phase. This will inevitably lead to its greater development in the medical field and this study represents only a primordial evaluation of what AI can offer as a decision supporting tool.

## Figures and Tables

**Figure 1 cancers-13-04635-f001:**
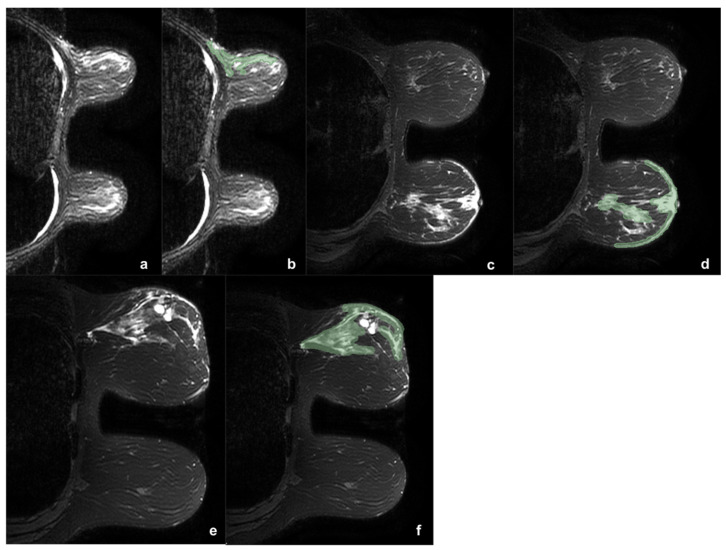
Three example of breast tumor associated edema before (**a**,**c**,**e**) and after (**b**,**d**,**f**) the segmentation process (green). In the first case (**a**,**b**) was reported a case of upper-external quadrant CDI, LUMINAL B, G2 tumor, with pre-pectoral and peritumoral edema. In the second case (**c**,**d**) was shown a case of central CDI, LUMINAL B, G3 tumor, with peritumoral and sub-cutaneous edema. It is interesting to note as in the second case the comparison with the contralateral breast helps in the edema definition. The third case (**e**,**f**) shows a CDI, LUMINAL B, G3 tumor, with diffuse edema. During the segmentation the two hyperintense areas with regular margins were identified as cysts and accurately avoided during the ROI definition.

**Figure 2 cancers-13-04635-f002:**
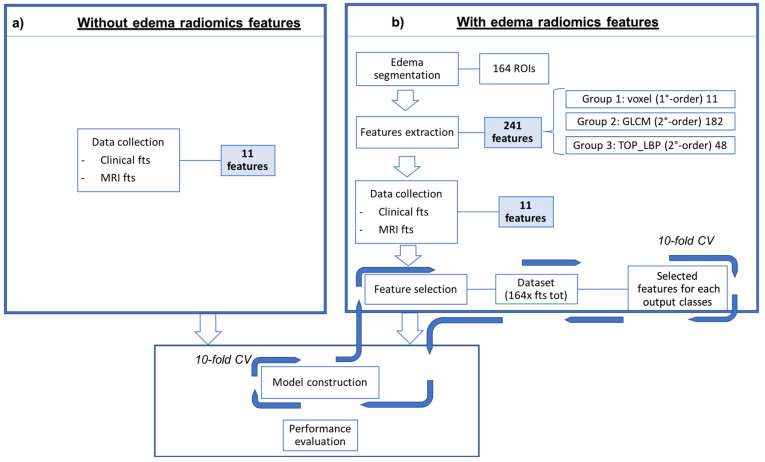
Schematic representation of the proposed approach to predict the five histological labels. In the left bow (**a**) is reported the classic medical approach, using only the semantic features, while in the right box (**b**) the classification was performed associating the radiomics features to the semantic ones. The model is then built using the best performing features.

**Figure 3 cancers-13-04635-f003:**
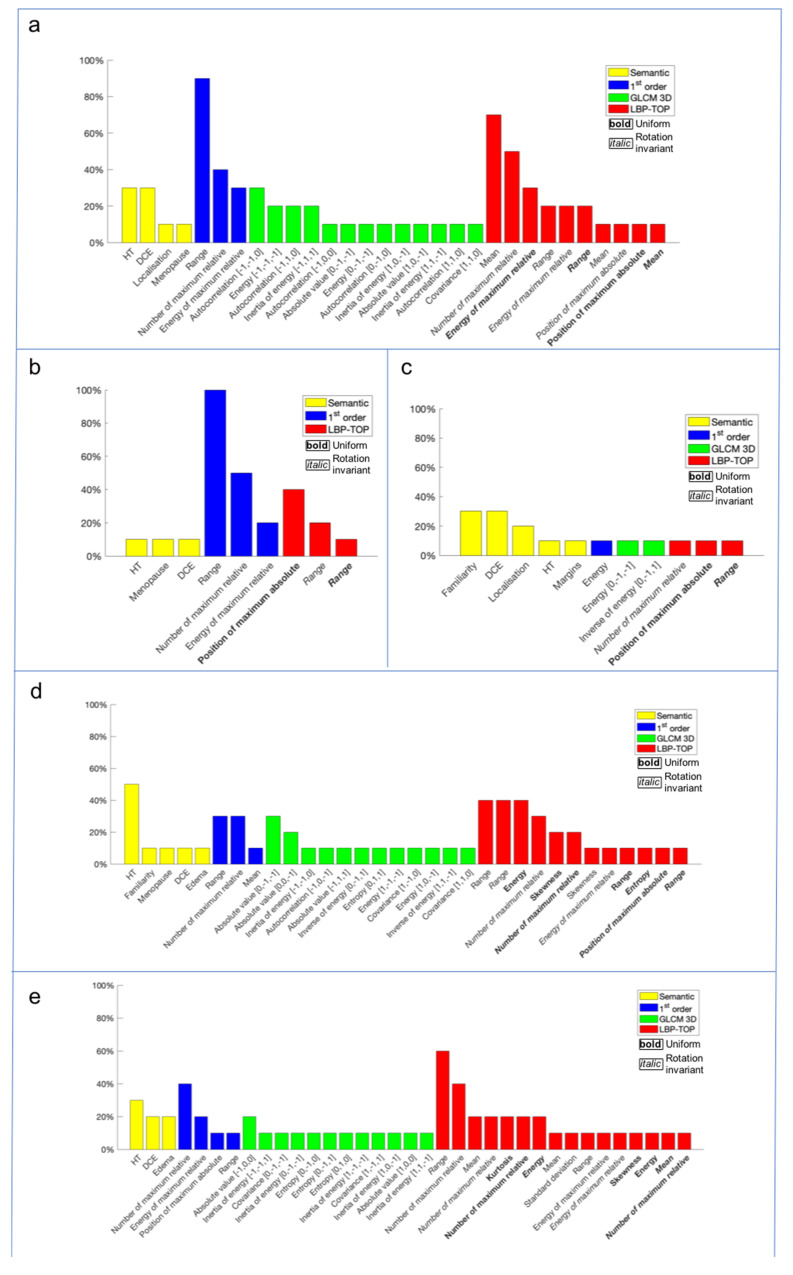
The charts show the most significant features, selected by a wrapper filter, for each histological label: (**a**) grading, (**b**) ER, (**c**) histological class, (**d**) Ki-67 and (**e**) PgR. The features were individuated by a color based on the specific feature class (yellow for semantic features, i.e., the 11 features collected without considering the edema information, blue for radiomics first-order features, green for radiomics GLCM 3D second-order features and red for radiomics LBP 3D second-order features).

**Table 1 cancers-13-04635-t001:** Description of the extracted features. #: “number of”.

	Features *n*	Parameters	Description
Semantic features	11	-	Clinical (age, HT, family history, menopausal state) and imaging (location, stadiation, margins, dimensions, morphology, kinetic curves, edema type)
First-order	12	# bins = 2^16^	All these features base on the count of the voxels in a ROI and therefore on the associated histogram computed on such count
Second-order TOP-LBP	48	Radius = 1 # neighbors = 8	These features attempt to extract the shape’s pattern of tumour inside a ROI analysing the neighborhood of each voxel
Second-order GLCM	182	Interpixel distance = 1	These are the multi-dimensional generalization of the histogram and aim to determine the tissue’s orientation inside a ROI

**Table 2 cancers-13-04635-t002:** Clinical, histologic, and MRI characteristics of the patients presenting with breast edema, classified according to the type of edema (* means the correlation is statistically significant). HT: hormone therapy, IDC: Invasive Ductal Carcinoma, ILC: Invasive Lobular Carcinoma, LNS: Lymph Node Status.

			Edema Type				Total	
			Peritumoral	Pre-Pectoral	Subcutaneous	Diffuse		*p*-Value
Family History	None	*n*	37	17	14	18	86	0.0250
		%	29.1%	13.4%	11.0%	14.2%	67.7%	
	1	*n*	18	8	2	4	32	
		%	14.2%	6.3%	1.6%	3.1%	25.2%	
	>1	*n*	7	1	0	1	9	
		%	5.5%	0.8%	0.0%	0.8%	7.1%	
Hormone Therapy	None	*n*	58	26	15	19	118	0.267
		%	45.7%	20.5%	11.8%	15.0%	92.9%	
	Positive	*n*	4	0	1	4	9	
		%	3.1%	0.0%	0.8%	3.1%	7.1%	
Menopause	Pre-m	*n*	24	14	5	6	49	0.444
		%	18.9%	11.0%	3.9%	4.7%	38.6%	
	Post-	*n*	38	12	11	17	78	
		%	29.9%	9.4%	8.7%	13.4%	61.4%	
Kinetic Curve	I	*n*	9	2	2	6	19	0.375
		%	7.1%	1.6%	1.6%	4.7%	15.0%	
	II	*n*	32	13	4	7	56	
		%	25.2%	10.2%	3.1%	5.5%	44.1%	
	III	*n*	21	11	10	10	52	
		%	16.5%	8.7%	7.9%	7.9%	40.9%	
Margins	Regular	*n*	3	0	1	0	4	0.746
		%	2.4%	0.0%	0.8%	0.0%	3.1%	
	Irregular	*n*	28	14	11	12	65	
		%	22.0%	11.0%	8.7%	9.4%	51.2%	
	Lobulated	*n*	7	5	0	2	14	
		%	5.5%	3.9%	0.0%	1.6%	11.0%	
	Spiculated	*n*	21	6	4	5	36	
		%	16.5%	4.7%	3.1%	3.9%	28.3%	
	Non-mass	*n*	3	1	0	4	8	
		%	2.4%	0.8%	0.0%	3.1%	6.3%	
Histology	IDC	*n*	54	23	16	17	110	0.513
		%	42.5%	18.1%	12.6%	13.4%	86.6%	
	ILC	*n*	8	3	0	6	17	
		%	6.3%	2.4%	0.0%	4.7%	13.4%	
Grade	1	*n*	11	1	0	2	14	<0.001 *
		%	8.7%	0.8%	0.0%	1.6%	11.0%	
	2	*n*	33	8	6	6	53	
		%	26.0%	6.3%	4.7%	4.7%	41.7%	
	3	*n*	18	17	10	15	60	
		%	14.2%	13.4%	7.9%	11.8%	47.2%	
LNS	Negative	*n*	53	23	14	14	104	0.064
		%	41.7%	18.1%	11.0%	11.0%	81.9%	
	Positive	*n*	9	3	2	9	23	
		%	7.1%	2.4%	1.6%	7.1%	18.1%	
ER Status	Negative	*n*	7	8	5	6	26	0.029 *
		%	5.5%	6.3%	3.9%	4.7%	20.5%	
	Positive	*n*	55	18	11	17	101	
		%	43.3%	14.2%	8.7%	13.4%	79.5%	
PR Status	Positive	*n*	16	11	8	11	46	0.018 *
		%	12.6%	8.7%	6.3%	8.7%	36.2%	
	Negative	*n*	46	15	8	12	81	
		%	36.2%	11.8%	6.3%	9.4%	63.8%	
HER2 Status	Negative	*n*	58	24	14	15		0.003 *
		%	45.7%	18.9%	11.0%	11.8%		
	Positive	*n*	4	2	2	8		
		%	3.1%	1.6%	1.6%	6.3%		
Ki-67	<20%	*n*	30	6	1	6	43	0.004 *
		%	23.6%	4.7%	0.8%	4.7%	33.9%	
	>20%	*n*	32	20	15	17	84	
		%	25.2%	15.7%	11.8%	13.4%	66.1%	

**Table 3 cancers-13-04635-t003:** The AUC and the accuracy for each histological label are obtained, at first, using only the semantic features (second column) and, then, adding the edema radiomics features (third column). In the last column is reported the difference between the two AUC/accuracies, indicating with * the cases in which edema radiomics features increased the result.

	Without Edema	With Edema	Difference
Histology	AUC: 0.520	AUC: 0.645	AUC: +0.125 *
	Accuracy: 85.8%	Accuracy: 64.17%	Accuracy: −21%
	Sensibility: 100%	Sensibility: 64.7%	Sensibility: −35.3%
	Specificity: 5.6%	Specificity: 61.1%	Specificity: +55.5% *
	PPV: 85.7%	PPV: 90.4%	PPV: +4.7% *
	NPV: 100%	NPV: 23.4%	NPV: −76.6%
Grading	AUC: 0.590	AUC: 0.789	AUC: +0.199 *
	Accuracy: 90%	Accuracy: 90.8%	Accuracy: +0.8% *
	Sensibility: 0%	Sensibility: 36.4%	Sensibility: +36.4% *
	Specificity: 100%	Specificity: 96.3%	Specificity: −3.7%
	PPV: 0%	PPV: 50%	PPV: +50% *
	NPV: 90.8%	NPV: 93.8%	NPV: +3% *
ER	AUC: 0.466	AUC: 0.487	AUC: +0.021 *
	Accuracy: 72.5%	Accuracy: 81.7%	Accuracy: +9.2% *
	Sensibility: 0%	Sensibility: 23.1%	Sensibility: +23.1% *
	Specificity: 92.6%	Specificity: 97.9%	Specificity: +5.3% *
	PPV: 0%	PPV: 75%	PPV: +75% *
	NPV: 77%	NPV: 82.1%	NPV: +5.1% *
PR	AUC: 0.546	AUC: 0.659	AUC: +0.113 *
	Accuracy: 55%	Accuracy: 61.7%	Accuracy: +6.7% *
	Sensibility: 35.4%	Sensibility: 54.2%	Sensibility: +18.8% *
	Specificity: 68.1%	Specificity: 66.7%	Specificity: −1.4%
	PPV: 42.5%	PPV: 52%	PPV: +9.5% *
	NPV: 61.3%	NPV: 68.6%	NPV: +7.3% *
Ki-67	AUC: 0.573	AUC: 0.621	AUC: +0.048 *
	Accuracy: 59.2%	Accuracy: 64.2%	Accuracy: +5% *
	Sensibility: 17.1%	Sensibility: 34.1%	Sensibility: +17% *
	Specificity: 81%	Specificity: 79.7%	Specificity: −1.3%
	PPV: 31.8%	PPV: 46.7%	PPV: +14.9% *
	NPV: 65.3%	NPV: 70%	NPV: +4.7% *

## Data Availability

The data presented in this study are available on request from the corresponding author.

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
