# Peer review of "The Impact of Tumor Edema on T2-Weighted 3T-MRI Invasive Breast Cancer Histological Characterization: A Pilot Radiomics Study"

_cancers, 2021, doi:10.3390/cancers13184635_

Round 1

Reviewer 1 Report

The discussion part of the manuscript is interesting and promising. The clinical data and statistical analysis can be implemented for appropriate diagnosis. The study looks impressive; however, the writing and presentation should improve substantially before acceptance. The selective comments are:

  1. The Simple Summary part needs to rewrite. For example, the second line of the Simple Summary, “In addition to the post-contrast sequences, considered gold-standard for tumor diagnosis, the pre-contrast ones enrich the characterization of the lesions,” is confusing.
  2. The manuscript looks like draft writing and submission. The authors could send it to other co-authors for reading and check lines, grammar and paragraphs before submission. Alternatively, the author should read the published manuscript in Cancer to finalize the manuscript. The Abstract and Introduction need to write again and then check before submission.
  3. Figure 3 can be split into two different figures. It will help to magnify the figures and readability.
  4. The conclusion section did not say anything about the preliminary results and how they could utilize the outcomes. So it should rewrite according to a standard published manuscript.
  5. The manuscript is belonging to clinical data analysis. The author should add the approval number or registration ID in the Institutional Review Board Statement.

Author Response

Response to Reviewer 1 Comments

The discussion part of the manuscript is interesting and promising. The clinical data and statistical analysis can be implemented for appropriate diagnosis. The study looks impressive; however, the writing and presentation should improve substantially before acceptance. The selective comments are:

  1. The Simple Summary part needs to rewrite. For example, the second line of the Simple Summary, “In addition to the post-contrast sequences, considered gold-standard for tumor diagnosis, the pre-contrast ones enrich the characterization of the lesions,” is confusing.

Thanks to the reviewer for the suggestion. The first part of the Simple Summary has been rewrite as follows:

“Breast cancer is the most common cancer in women worldwide. Currently the use of MR is mandatory in staging phase. The standard protocol includes T2-weighted mandatory in staging phase. The standard protocol includes T2-weighted sequences for anatomy and signal analysis, T1-weighted images for adding information (i.e. ematic or adipous component), diffusion-weighted sequences which provide information about tissue cellularity and dynamic post-contrast sequences useful for detection and locating lesions. Although not considered among the main prognostic factors in current guidelines, tumor-associated edema provides useful information on tumor aggressiveness, and has been shown to be associated with the main histological tumor characteristics.”

  1. The manuscript looks like draft writing and submission. The authors could send it to other co-authors for reading and check lines, grammar and paragraphs before submission. Alternatively, the author should read the published manuscript in Cancer to finalize the manuscript. The Abstract and Introduction need to write again and then check before submission.

We thank the reviewer for these indications helping us improving the comprehensibility of the work. The manuscript was reviewed in its entirety, particularly the Introduction and Abstract, and spell checks, grammatical corrections, and text clarifications were performed as requested.

  1. Figure 3 can be split into two different figures. It will help to magnify the figures and readability.

The Figure 3 has been changed in format and size helping the magnification.

  1. The conclusion section did not say anything about the preliminary results and how they could utilize the outcomes. So it should rewrite according to a standard published manuscript.

Following reviewer’s indications the following paragraph has been added in the Conclusion section:

“Despite its currently marginal role in guidelines as a predictor, the breast associated edema may be useful in the preliminary phase, to provide additional information on tumor bio-histological aggressiveness. This concept is more evident by applying Artificial Intelligence algorithms to images that can provide objective quantitative data to the limited vision of the radiologist. Radiomics is a complex tool in its implementation phase but simple in its use phase. This will inevitably lead to its greater development in the medical field and this study represents only a primordial evaluation of what AI can offer as a decision supporting tool.”

  1. The manuscript is belonging to clinical data analysis. The author should add the approval number or registration ID in the Institutional Review Board Statement.

ID ethical registration protocol number: 24.21 (OSS.NOT)

Reviewer 2 Report

This paper describes radiomics analysis of tumor edema for the purposes of breast tumor histological classification. The paper is reasonably well written and relatively concise. The subject is of interest but there are a number of points regarding the analysis that need to be addressed. These include justification for the use of home-made software, a more detailed description of the texture analysis pipeline, further parameter reduction prior to model development (or justification for the use of so many parameters), more complete results tables (including full diagnostic metrics), and more rigorous statistical analysis when comparing models with and without edema information.

Abstract

OK

Introduction

OK

Materials and Methods

Page 5 Lines 187-188 – Pearson’s chi squared test does not assess correlations, rather it tests the relationship between two categorical variables.

Page 5 Lines 205-206 – There are many well established packages that enable texture analysis. What is the justification for using home-made algorithms? Also, has the home-made software been tested against any established package or the image biomarker standardization initiative.

Page 5 Lines 205-206 – Currently there is not enough information present for readers to attempt to replicate the texture analysis. So, for example, what GLCM features were calculated and for what interpixel distances? Was the data decimated/preprocessed to reduce the number of gray levels and deal with outlying image intensities before analysis. Were features calculated over multiple 2D slices or in a 3D fashion? 3D doesn’t seem appropriate because of the relatively large slice thickness compared to the in-plane resolution. Please provide more information here.

Page 5 Lines 209-210 – I’m not sure I understand why a convex hull algorithm is useful here, please clarify. Surely the edema is not confined to a convex hull, and can contain concavities in its outline?

Results

Page 7 Lines 254-256 – The abstract states that there are 160 patients in this study, so with 5 patients presenting with bilateral cancer there should be 165 lesions, rather than 164 as suggested here?

Page 7 Lines 257-263 – It is incorrect to present the pre/post-menopausal, family history and hormone therapy data in this manner. There are 160 patients, so the totals must match this (not 164 for menopausal and hormone therapy, and rather strangely 163 for family history). The authors are using a lesion-based breakdown rather than a patient-based breakdown here and are thus incorrectly double counting some patients.

Pages 8-9 Table 2 – The first line of the results section states that the number of lesions presenting edema is 126, so why do most of these categories sum to 127? In fact, they all sum to 127 apart from kinetic curve type (126) and histology (125). Please thoroughly check this data.

Page 9 Lines 281-286 – Again, the Pearson chi squared test is not assessing correlations and references to this should be removed.

Page 9 Figure 3 – From this figure the radiomics based models all seem to be overparameterized. As a rule of thumb there should be a minimum of 5 cases (ideally at least 10) per parameter for the minority class. For grade there are 30 parameters (minority class size 60), for ER status there are 9 parameters (minority class size 26), for histology type there are 11 parameters (minority class size 17), for PR status there are 35 parameters (minority class size 46) and for Ki-67 status there are 32 parameters (minority class size 43). The authors need to consider redeveloping models after further feature reduction.

Page 9 Figure 3 – Please clarify how many parameters are utilized in developing models without edema information. Presumably just those represented by the yellow bars?

Pages 9-10 Table 3 – This table is not very informative. It would be more appropriate to include more complete information for the predictive capabilities of each model i.e. sensitivity. specificity, NPV and PPV data. This is especially important because of the large class imbalances. So, the histology model without edema information has an accuracy of 85.8% which initially appears to be very good but with 101/127 cases of IDC this accuracy can be achieved by assigning all cases to this class!

Pages 9-10 Table 3 – The final column of this table demonstrating differences in AUC and accuracy between the non-edema and edema based models is not very informative unless the data is reinforced with statistical analysis. So, the accuracies can be compared with the McNemar test and the AUC values with the deLong test. These need to be performed to demonstrate the efficacy of utilizing radiomic features from the edema. Otherwise the results are merely anecdotal.

Discussion

A paragraph on study limitations needs to be added here. These include, but are not limited to, small sample size (especially in the minority classes), the single center nature of the study, the large number of features utilized leading to overparameterization, the lack of an independent test set, and the lack of an assessment of inter-reader variation. The authors need to make it clear that the study is really hypothesis generation and would need to be explored further in a larger, preferably multi-center, cohort.

Author Response

Response to Reviewer 2 Comments

This paper describes radiomics analysis of tumor edema for the purposes of breast tumor histological classification. The paper is reasonably well written and relatively concise. The subject is of interest but there are a number of points regarding the analysis that need to be addressed. These include justification for the use of home-made software, a more detailed description of the texture analysis pipeline, further parameter reduction prior to model development (or justification for the use of so many parameters), more complete results tables (including full diagnostic metrics), and more rigorous statistical analysis when comparing models with and without edema information.

Abstract

OK

Introduction

OK

Materials and Methods

Page 5 Lines 187-188 – Pearson’s chi squared test does not assess correlations, rather it tests the relationship between two categorical variables.
Thanks to the reviewer for this crucial suggestion. Correlations were recalculated using the Spearman correlation test. All p values reported in the text and Table 1 were modified accordingly. We added a correlation between ADC values and classes of edema, excluding peritumoral edema.

Page 5 Lines 205-206 – There are many well established packages that enable texture analysis. What is the justification for using home-made algorithms? Also, has the home-made software been tested against any established package or the image biomarker standardization initiative.
We would like to thank the reviewer for the opportunity to further clarify this aspect. Although there are several well established packages that allow the partial implementation of the proposed Machine Learning pipeline, we preferred to compute the extraction of image features using home-made software. The rationale behind this choice is based on the fact that not all the features considered are naturally extensible from a 2D to a 3D space, therefore a manual implementation of the algorithms to compute them allowed us to have a finer control on specific aspects of the code, e.g. the management of non-ROI voxels and the eventual data imputation on the edge of a ROI. However, it is worth noting that all the steps required to build our codes were previously validated against any established packages. We have addressed this point with a footnote in the revised manuscript.

Page 5 Lines 205-206 – Currently there is not enough information present for readers to attempt to replicate the texture analysis. So, for example, what GLCM features were calculated and for what interpixel distances? Was the data decimated/preprocessed to reduce the number of gray levels and deal with outlying image intensities before analysis. Were features calculated over multiple 2D slices or in a 3D fashion? 3D doesn’t seem appropriate because of the relatively large slice thickness compared to the in-plane resolution. Please provide more information here.
We thank the reviewer for these observations to help us improving the comprehensibility of the work. The revised text is enriched in different points dealing with the given objections. Such changes are as follows:

  • We listed, inside the text, before the Table 1, the name of all the features we extracted from each feature group;
  • We added the column “Parameters” to Table 1 that lists the computational parameter used in the feature extraction when possible;
  • We explained the absence of the data and gray level reduction with the statement: “In order to fully take advantage of gathered acquisitions no data nor gray level reduction was performed to the images and all analysis were intrinsically computed in the three-dimensional voxel space.”;
  • We justified the 3D analysis with the statement: “It worth noting that although the relatively large slice thickness of the MRs could alter the isotropy of the acquired stacks, all the features we extracted which are sensitive to the voxels texture orientation were computed for all the existing three-dimensional orientations and then filtered out, in the further feature selection step, all the less informative ones.”.

Page 5 Lines 209-210 – I’m not sure I understand why a convex hull algorithm is useful here, please clarify. Surely the edema is not confined to a convex hull, and can contain concavities in its outline?
We thank the reviewer for pointing out the lack of description of the calculation of the convex hull. We have improved the manuscript by adding the following considerations at the end of subsection 2.6.1: “Indeed, although all tumor edemas usually present a spherical and irregular geometry, when considering an MR acquisition it is easy to incur in some tissues that are partially difficult to segment, either because of tissue stretching due to the imaging phase, or because of hard-to-detect shapes within the images when edemas are close to other tissues with similar density, such as cysts or pure dense glandular tissue, thus often resulting in jagged edges and sometimes concavities in the resulting ROI. On this ground, the calculation of the three-dimensional convex hull serves a twofold purpose: firstly, it removes all possible segmentation deformities otherwise hindering further calculation of textural features and secondly, it can slightly expand the ROIs thus also containing a perimeter portion of healthy tissue at the edge of the segmentation that might actually carry information about the tumor status.”.

Results

Page 7 Lines 254-256 – The abstract states that there are 160 patients in this study, so with 5 patients presenting with bilateral cancer there should be 165 lesions, rather than 164 as suggested here?
We thanks the reviewer. There were 4 patients with bilateral tumors, not 5. The text has been corrected accordingly.

Page 7 Lines 257-263 – It is incorrect to present the pre/post-menopausal, family history and hormone therapy data in this manner. There are 160 patients, so the totals must match this (not 164 for menopausal and hormone therapy, and rather strangely 163 for family history). The authors are using a lesion-based breakdown rather than a patient-based breakdown here and are thus incorrectly double counting some patients.
Accordingly to the reviewer suggestion, the text was modified to divide patients, not individual lesions, according to their medical history.

Pages 8-9 Table 2 – The first line of the results section states that the number of lesions presenting edema is 126, so why do most of these categories sum to 127? In fact, they all sum to 127 apart from kinetic curve type (126) and histology (125). Please thoroughly check this data.
The date was rechecked, and the number of lesions presenting edema corresponded to 127. The text has been modified, as has Table 1.

Page 9 Lines 281-286 – Again, the Pearson chi squared test is not assessing correlations and references to this should be removed.
As previously, correlations were recalculated using the Spearman correlation test. All p values reported in the text and Table 1 were modified accordingly.

Page 9 Figure 3 – From this figure the radiomics based models all seem to be overparameterized. As a rule of thumb there should be a minimum of 5 cases (ideally at least 10) per parameter for the minority class. For grade there are 30 parameters (minority class size 60), for ER status there are 9 parameters (minority class size 26), for histology type there are 11 parameters (minority class size 17), for PR status there are 35 parameters (minority class size 46) and for Ki-67 status there are 32 parameters (minority class size 43). The authors need to consider redeveloping models after further feature reduction.
We would like to thank the revisor for allowing us to address this point. After Figure 3 we inserted as follows: “For the sake of completeness, we would like to stress the fact that in order not to incur in the issue known as Curse of Dimensionality, i.e. when during a classification task the number of features is empirically higher than 10% of the samples present in the dataset, thus biasing the predictions, after the feature selection phase we should have concluded with at least 10 edemas in the minority class per feature, and instead the number of selected features exceeds the number of analysed samples in all the classes considered. In fact, after the feature selection we accepted a certain abundance of extra features without proceeding with a deeper filtering due to the fact that the algorithm considered in the next step, the Random Forest, is quite capable of dealing with a larger number of attributes than the relevant ones, performing an additional intrinsic selection of only the most significant features and avoiding bias.”.

Page 9 Figure 3 – Please clarify how many parameters are utilized in developing models without edema information. Presumably just those represented by the yellow bars?
We thank the reviewer. We have updated the caption of Figure 3 by specifying the number and nature of features considered without the edema information.

Pages 9-10 Table 3 – This table is not very informative. It would be more appropriate to include more complete information for the predictive capabilities of each model i.e. sensitivity. specificity, NPV and PPV data. This is especially important because of the large class imbalances. So, the histology model without edema information has an accuracy of 85.8% which initially appears to be very good but with 101/127 cases of IDC this accuracy can be achieved by assigning all cases to this class!
Thanks to the reviewer’s observations we have expanded the Table 3 with Sensitivity, Specificity, PPV and PPN values and added the following statement in the middle of the Discussion section when describing the corresponding results: “The improvements can be confirmed by looking at the values of sensitivity, specificity, PPV and NPV of each experiment. In fact, the percentages almost always show higher values when considering predictions obtained using edemas when calculating features rather than not, only in a few comparisons we have that specificity shows a lower result as well as a single NPV, suggesting a more conservative trend for the classifier using edema features.”.

Pages 9-10 Table 3 – The final column of this table demonstrating differences in AUC and accuracy between the non-edema and edema based models is not very informative unless the data is reinforced with statistical analysis. So, the accuracies can be compared with the McNemar test and the AUC values with the deLong test. These need to be performed to demonstrate the efficacy of utilizing radiomic features from the edema. Otherwise the results are merely anecdotal.

Discussion

A paragraph on study limitations needs to be added here. These include, but are not limited to, small sample size (especially in the minority classes), the single center nature of the study, the large number of features utilized leading to overparameterization, the lack of an independent test set, and the lack of an assessment of inter-reader variation. The authors need to make it clear that the study is really hypothesis generation and would need to be explored further in a larger, preferably multi-center, cohort.
Thanks for this clarification. The limitation paragraph was rewritten as follows: “The main limitations of this study include the small sample size and the single center nature of the study, the unbalanced number of the groups component (smaller number of the minority class), the manual segmentation, leading to time-consumption and user error and variability, the large number of features utilized which may lead to overparameterization, the lack of an independent test set, and the lack of an assessment of inter-reader variation. A larger and preferably multi-center cohort is needed for a more rigorous analysis. Evenmore, an automatic, stadardized, and validated segmentation method would be ideal even if not yet available at present. It is certainly necessary to keep in mind the inherent limits of radiomics: this technique is currently still tied to the radiologist's ability to select ROIs and the engineer's ability to build the algorithm capable of feature selection and their experience”.

Reviewer 3 Report

The manuscript provides the demonstration of radiomics edema and enhancement of the accuracy in the prediction of histological factors, which is significantly of interest. There seems to, however, be a lack of detail in some important aspects.

(1) P. 1, l.39

The authors mentioned that “The AUC-classifier was always higher using edema radiomics …”. What was the AUC-classifier higher compared with?

(2) P. 3, l.123

Why did the authors perform five times of MRI sequences after contrast administration? To obtain the time series, the authors did that? Could the authors explain the purpose of five times of MRI?

(3) P. 4, Figure 1

The authors displayed the case with pre-pectoral, peritumoral, and sub-cutaneous edema in Fig. 1. In this manuscript, there are four classifications of edema. Could you also display the case of “diffuse”?

(4) P. 9, Figure 9

It is hard to see the characters in Fig. 9. Could you enlarge the characters and re-organize the materials for easy reading?

Minor comments;

P. 12, l. 422

It seems that there is an unnecessary space between the words “correlate” and “with”.

Author Response

Response to Reviewer 3 Comments

The manuscript provides the demonstration of radiomics edema and enhancement of the accuracy in the prediction of histological factors, which is significantly of interest. There seems to, however, be a lack of detail in some important aspects.

(1) P. 1, l.39

The authors mentioned that “The AUC-classifier was always higher using edema radiomics …”. What was the AUC-classifier higher compared with?
The sentence has been changed and clarified as suggested by the reviewer:
“The AUC-classifier obtained associating the edema radiomics to semantic features was always higher compared to the AUC-classifier obtained from semantic features alone, for all the five histological classes prediction” 

(2) P. 3, l.123

Why did the authors perform five times of MRI sequences after contrast administration? To obtain the time series, the authors did that? Could the authors explain the purpose of five times of MRI?
Thanks to the reviewer to this suggestion that allows us to add the following expaination:
“Current guidelines suggest at least three time points to measure during the post-contrast-phase: one before the administration of contrast medium, one approximately 2 min later to capture the peak, one in the late phase. This allows us to evaluate whether a lesion continues to enhance or is characterized by contrast agent wash-out. The performance of at least two measurements after the contrast medium is recommended, even if the optimal number of repetitions is unknown. In our center, we use to perform 5 acquisitions after contrast medium administration ensuring to obtain a specific signal intensity curve time without penalizing the duration of the exam.”

(3) P. 4, Figure 1

The authors displayed the case with pre-pectoral, peritumoral, and sub-cutaneous edema in Fig. 1. In this manuscript, there are four classifications of edema. Could you also display the case of “diffuse”?
Thanks to this suggestion we enrich the Fig1 with the example of diffuse edema

(4) P. 9, Figure 9

It is hard to see the characters in Fig. 9. Could you enlarge the characters and re-organize the materials for easy reading?
We thank the reviewer for the observation. In the revised manuscript we listed, inside the text, before the Table 1, the name of all the features we extracted from each feature group.
Maybe the Reviewer refers to Figure 3. The Figure 3 has been changed in format and size helping the magnification. In the Materials a paragraph regarding the specific features for each class has been added in the 2.6.2 subsection.

Minor comments;

  1. 12, l. 422

It seems that there is an unnecessary space between the words “correlate” and “with”. The error has been corrected

Round 2

Reviewer 1 Report

The authors have made all the necessary corrections. The editor could accept the current version of the manuscript.

Author Response

Thank you for the revisions which helped us to improve our work

Reviewer 2 Report

The authors have appropriately addressed the points raised in my initial review. 

Author Response

Thanks to the revisor who helped us to improve our work